# Diagnostic and Prognostic Roles of Urine Nectin-2 and Nectin-4 in Human Bladder Cancer

**DOI:** 10.3390/cancers15092565

**Published:** 2023-04-29

**Authors:** Makito Miyake, Nobutaka Nishimura, Sayuri Ohnishi, Yuki Oda, Takuya Owari, Kenta Ohnishi, Yosuke Morizawa, Shunta Hori, Daisuke Gotoh, Yasushi Nakai, Kazumasa Torimoto, Tomomi Fujii, Nobumichi Tanaka, Kiyohide Fujimoto

**Affiliations:** 1Department of Urology, Nara Medical University, 840 Shijo-cho, Kashihara 634-8522, Nara, Japan; ffxxxx.nqou@gmail.com (N.N.); sayuri3@naramed-u.ac.jp (S.O.); tintherye@gmail.com (T.O.); kenzmedico0912@yahoo.co.jp (K.O.); tigers.yosuke@gmail.com (Y.M.); horimaus@gmail.com (S.H.); dgotou@gmail.com (D.G.); nakaiyasusiuro@live.jp (Y.N.); torimoto@naramed-u.ac.jp (K.T.); sendo@naramed-u.ac.jp (N.T.); kiyokun@naramed-u.ac.jp (K.F.); 2Department of Diagnostic Pathology, Nara Medical University, 840 Shijo-cho, Kashihara 634-8522, Nara, Japan; fujiit@naramed-u.ac.jp; 3Department of Prostate Brachytherapy, Nara Medical University, 840 Shijo-cho, Kashihara 634-8522, Nara, Japan

**Keywords:** Nectin, NMP-22, cytology, bladder cancer, urine, serum, expression, biomarker, detection

## Abstract

**Simple Summary:**

The clinical utility of urine nectins in bladder cancer (BCa) is unclear. We investigated the potential diagnostic and prognostic values of Nectin-2 and Nectin-4. This study included 122 patients with BCa, including 78 with non-muscle-invasive BCa, 44 with muscle-invasive BCa, and ten healthy controls. The detection sensitivities of urine Nectin-2, urine Nectin-4, NMP-22, and cytology were 84%, 98%, 52%, and 47%, respectively. Their specificities were 40%, 80%, 100%, and 100%, respectively. Urine Nectin-2 and Nectin-4, but not NMP-22, were significantly more sensitive than cytology alone. A four-titer grouping based on levels of urine Nectin-2/Nectin-4 had a high capability for discriminating between NMIBC and MIBC. Urine levels correlated with tumor expression and serum levels in the Nectin-4 analysis. Urine nectins are potential diagnostic biomarkers for BCa.

**Abstract:**

The clinical utility of urine nectins in bladder cancer (BCa) is unclear. We investigated the potential diagnostic and prognostic values of urine Nectin-2 and Nectin-4. Levels of urine Nectin-2, Nectin-4, and NMP-22 were quantified using an enzyme-linked immunosorbent assay in 122 patients with BCa, consisting of 78 with non-muscle-invasive BCa (NMIBC) and 44 with muscle-invasive BCa (MIBC), and ten healthy controls. Tumor nectin expression in MIBC was evaluated with immunohistochemical staining of transurethral resection specimens. The level of urine Nectin-4 (mean: 18.3 ng/mL) was much higher than that of urine Nectin-2 (mean: 0.40 ng/mL). The sensitivities of Nectin-2, Nectin-4, NMP-22, and cytology assays were 84%, 98%, 52%, and 47%, respectively; their specificities were 40%, 80%, 100%, and 100%, respectively. Both urine Nectin-2 and Nectin-4, though not NMP-22, were found to be significantly more sensitive than cytology. A four-titer grouping based on levels of urine Nectin-2/Nectin-4 (low/high, high/high, low/low, and high/low) showed a high capability for discriminating between NMIBC and MIBC. Neither urine Nectin-2 nor Nectin-4 levels had a significant prognostic value in NMIBC or MIBC. Urine levels correlated with tumor expression and serum levels in the Nectin-4 analysis, but not in the Nectin-2 analysis. Urine nectins are potential diagnostic biomarkers for BCa.

## 1. Introduction

Combinations of urine cytology, cystoscopy, and conventional radiographic examinations, including computed tomography and magnetic resonance imaging, are currently used to diagnose and monitor bladder cancer (BCa). Voided urinary cytology (VUC) is the most widely used non-invasive urine test, with poor sensitivities ranging from 16% to 53% and high specificities ranging from 94% to 100% [1]. Owing to the poor cost-effectiveness of cystoscopy and imaging tests, there is an urgent need for inexpensive non-invasive urine-based tests for health checkups, hematuria screening, and postsurgical surveillance for urothelial carcinoma (UC).

The United States Food and Drug Administration, European Medicines Agency, and Pharmaceuticals and Medical Devices Agency in Japan have approved a Nectin-4-directed antibody conjugated to the microtubule-disrupting cytotoxic agent monomethyl auristatin E, enfortumab vedotin (EV; Padcev^®^), for patients with unresectable/metastatic UC who have been previously been treated with chemotherapeutic agents and programmed cell death 1/programmed cell death 1 ligand 1 inhibitors, based on the results of the EV-301 trials [2]. The nectin family comprises four nectins, nectin 1–4, which resemble type I integral membrane proteins with ectodomains composed of three immunoglobulin domains. Nectins are mainly involved in Ca++-independent cell–cell adhesion by forming tight junctions via homophilic and heterophilic trans-interactions [3]. There is a close relationship between nectins and the actin cytoskeleton through afadin, and these complexes with adjacent cells regulate various cellular events, including cellular adhesion, movement, and polarization [4,5,6]. In normal organs, Nectin-1 and Nectin-2 are predominantly associated with immune system organs; nectin-3 is mainly expressed in the testes and placenta, whereas Nectin-4 is primarily expressed in the embryo and placenta [7,8,9].

Based on the evidence of Nectin-2 and Nectin-4 overexpression in solid malignancies, several studies have investigated their potential diagnostic and prognostic roles as serum biomarkers in ovarian [10,11], breast [12], lung [13,14], and colorectal cancers [15]. Our recent study demonstrated that nectins are predominantly expressed in the membrane and/or cytoplasm of UC cells with no or faint expression of Nectin-1, slight expression of Nectin-3, and high expression of Nectin-2 and Nectin-4 [16]. To the best of our knowledge, no study has addressed the potential diagnostic and prognostic values of urine Nectin-2 and Nectin-4 in any malignancy. We hypothesized that urine Nectin-2 and Nectin-4 were significantly higher in patients with BCa as compared to healthy controls. Here, we quantified the urine levels of Nectin-2 and Nectin-4 using enzyme-linked immunosorbent assay (ELISA) in patients with non-muscle-invasive BCa (NMIBC) and muscle-invasive bladder cancer (MIBC). In addition, we evaluated the correlations between urine nectin, serum nectin, and tumor nectin expression. The aim of this study was to evaluate whether these two nectins have potential as urine-based detection markers, as well as prognostic markers for BCa in comparison with other existing urine tests, including VUC and nuclear matrix protein-22 (NMP-22).

## 2. Patients and Methods

### 2.1. Patient Selection and Data Collection

This retrospective study included patients with BCa who were diagnosed between January 2010 and December 2021 and treated with transurethral resection of the bladder tumor (TURBT) and/or radical cystectomy with or without perioperative systemic chemotherapy at Nara Medical University Hospital. The inclusion criteria of this study were (1) pre-TURBT voided urine was preserved properly and available for ELISA (for NMIBC), (2) all of the pre-TURBT voided urine, pre-TURBT serum, and paraffin-embedded, formalin-fixed tissue blocks of TURBT specimens were preserved properly and available for the analysis (for NMIBC), and (3) written informed consent was obtained. A total of 122 pre-TURBT-voided urine samples from 78 patients with NMIBC and 44 patients with MIBC and ten healthy controls were analyzed by ELISA. Pre-TURBT serum samples and paraffin-embedded formalin-fixed tissues of TURBT specimens from 44 patients with MIBC were subjected to ELISA and Immunohistochemical (IHC) staining analyses, respectively (Figure 1A). Clinicopathological information and follow-up data were collected through a retrospective review of the patients’ medical charts. Follow-up was performed according to the institutional protocol [17,18]. The age of the healthy controls ranged from 46 to 65 years, and those consisted of seven males and three females. All controls underwent routine medical checkups once a year and revealed no evidence of malignant disease, including urogenital cancer or chronic kidney disease, which could cause proteinuria.

### 2.2. Processing of Voided Urine and Serum Samples

In total, 50 mL of fresh voided urine samples were obtained from patients immediately before TURBT and from control participants. Of these, 20 mL was sent to the pathological department for VUC examination by the Papanicolaou procedure [19], and the remaining sample was centrifuged at 400× *g* for 5 min at 20 °C. The supernatant was decanted, and 2 mL aliquots were prepared with urine stabilizer containing 2 mM Tris-HCL (pH 7.6), 0.1% bovine serum albumin, 0.09% sodium azide, and a protease inhibitor cocktail (Cat# P8340, Sigma-Aldrich, St. Louis, MO, USA). Peripheral blood was collected from patients with MIBC before TURBT into tubes with a serum separator and centrifuged at 1000× *g* for 15 min to obtain the supernatant (serum). Both the urine supernatant and serum samples were stored at −80 °C until analysis.

### 2.3. Measurement of Nectins in Urine and Serum

The levels of nectin and NMP-22 were measured using commercially available ELISA kits (Human Nectin 2 ELISA Kit; cat no. HUFI08699; Assay Genie, Dublin, Ireland; Human Nectin-4 Quantikine ELISA Kit, cat. DNEC40, R&D Systems, Minneapolis, MN, USA; Alere NMP22^®^ Bladder Cancer ELISA Test, Alere Scarborough, Inc., Waltham, MA, USA) according to the manufacturer’s protocol directions. Standard curves for nectin were generated using reference concentrations provided by the kit. Absorbance was measured following color development using a microplate spectrophotometer (Infinite 200 M PRO, Tecan, Männedorf, Switzerland) equipped with i-control version 1.8 software. The levels of Nectin-2, Nectin-4, and NMP-22 were expressed as ng/mL, ng/mL, and U/mL, respectively. To avoid experimental inconsistency among assays, all the ELISAs were performed at the same time.

The NMP-22 assays are FDA approved for the detection and surveillance of BCa in urine samples. For BCa detection, urinary NMP-22 tests have diagnostic sensitivities ranging from 47–100% and specificities ranging from 55–98% [20,21,22].

### 2.4. Immunohistochemical (IHC) Staining Analysis of Tumor Expression of Nectins

IHC staining was performed using paraffin-embedded, formalin-fixed tissue blocks of TURBT specimens, as previously described [16]. Primary antibodies (Atlas Antibodies, Stockholm, Sweden) used in this study were anti-Nectin-2 antibody (cat no. HPA0112759, dilution 1:200; Sigma-Aldrich, St. Louis, MO, USA) and anti-Nectin-4 antibody (cat no. HPA0112759, dilution 1:200; Sigma-Aldrich). The membrane and cytoplasmic expression of nectins was evaluated in at least three independent high-power microscopic fields of UC cells, which were determined based on cell morphology and tumor architecture. The percentage of positively stained UC cells divided by the total number of UC cells was calculated (1–100%). The intensity and extent of tumor nectin expression were determined using the histochemical scoring method (H-score) (Figure 1B; score of 0–3), which was calculated as the product of the staining intensity multiplied by the percentage of cells (0–100%) stained at a given intensity [3]. The IHC staining results were evaluated by two investigators (N. Nishimura and Y. Oda) who were blinded to the clinicopathological data.

### 2.5. Statistical Analysis

The two-sided Mann–Whitney *U* test or Kruskal–Wallis test was applied to evaluate differences in the levels of urine nectins between BCa patients and controls with respect to muscle invasiveness and T category. Urine nectin data were visualized as box-and-whisker plots and tabulated as the median, interquartile range (IQR), and mean ± standard error of the mean. The diagnostic performance of urine Nectin-2 and Nectin-4 was determined using receiver operating characteristic (ROC) curve analyses and area under the ROC curve (AUROC) values, followed by determination of the optimal cut-off value by calculating the Youden index [23]. The sensitivities of urine nectin and VUC were compared using McNemar’s test. Based on the published literature, a cutoff value of 10 U/mL was used to define a positive NMP-22 test [21,22]. The cut-offs for the prognostic performance of urine Nectin-2 and Nectin-4 levels were determined using the median level. In the NMIBC cohort, recurrence-free survival (RFS), progression-free survival (PFS), and overall survival (OS) from the day of TURBT were obtained using the Kaplan–Meier method and compared using the log-rank test. Recurrence was defined as recurrent tumors of pathologically proven urothelial carcinoma in bladder and prostatic urethra. Progression was defined as recurrent disease with invasion into the muscularis propria (≥T2), positive regional lymph nodes, and/or distant metastases. In the cohort of MIBC, disease-free survival (DFS), cancer-specific survival (CSS), and OS from the day of radical cystectomy were evaluated. DFS was defined as survival without any of urothelial tract recurrence, unresectable lesion, and distant metastasis. The correlations between parameters were examined using the Spearman correlation coefficient (*ρ* value) and linear regression analysis (Y-slope). The absolute values of Spearman *ρ* < 0.2–0.4 were considered to indicate weak correlation; 0.4–0.7, moderate correlation; and >0.7, strong correlation.

The PRISM software version 9.5.1 (GraphPad Software Inc., La Jolla, CA, USA) was used for statistical analyses and data plotting. For all analyses, two-sided *p <* 0.05 was considered statistically significant.

## 3. Results

### 3.1. Levels of Urine Nectin-2 and Nectin-4

The patient demographics are shown in Table 1. We compared the levels of urinary Nectin-2 and Nectin-4 in ten healthy control individuals, 78 patients with NMIBC, and 44 patients with MIBC. Generally, the level of urine Nectin-4 (mean: 18.3 ng/mL) was much higher than that of urine Nectin-2 (mean: 0.40 ng/mL) in patients with BCa. The mean ± standard error of the mean of urine Nectin-2 and Nectin-4 in the control group were 0.63 ± 0.34 ng/mL and 1.68 ± 1.01 ng/mL, respectively. Urine Nectin-2 levels did not differ between the BCa and control groups, while a higher T stage was associated with higher levels of urine Nectin-2 (Figure 2A,B). In contrast, urine levels of Nectin-4 were significantly higher in the BCa group than in the control group, and a higher T stage was associated with a lower level of urine Nectin-4 (Figure 2C,D). The media/mean values of urine nectin levels and their associations with baseline characteristics are listed in Table 1. Positivity for urine NMP-22 was associated with higher levels of Nectin-2, though not Nectin-4. High tumor grade and muscle invasiveness (≥T2 tumors) were strongly associated with lower levels of urine Nectin-4 (*p* < 0.001).

### 3.2. Diagnostic Performance of Urine Nectin-2 and Nectin-4

We performed ROC analyses to evaluate the diagnostic performance of urine Nectin-2 and Nectin-4 levels in discriminating between patients with BCa and controls. ROC curve analysis of the model allowed for the definition of cutoff values for each nectin (Figure 2E,G). Table 2 summarizes the AUROC, optimal cutoff values, and diagnostic performances. The sensitivities for Nectin-2, Nectin-4, NMP-22, and VUC were 84%, 98%, 52%, and 47%, respectively. Their specificities were 40%, 80%, 100%, and 100%, respectively. The urine Nectin-4 had the greatest diagnostic accuracy (96%), in which 119 of 122 patients (98%) with BCa were positive and 2 of 10 controls (20%) were positive. Using the McNemar’s test, both urine Nectin-2 and Nectin-4, not NMP-22, were found to be more sensitive than VUC as the reference standard for BCa detection.

The urine Nectin-2 was higher in MIBC than in NMIBC, whereas the urine Nectin-4 was higher in NMIBC than in MIBC (Figure 1A,C). To evaluate the potential of urine nectins as diagnostic markers to distinguish patients with NMIBC from those with MIBC, we performed ROC analyses and defined specific cutoff values for each nectin (Figure 2F,H). The optimal cutoff values were 0.011 ng/mL for Nectin-2 and 12.0 ng/mL for Nectin-4, providing 61% and 91% sensitivity and 61% and 86% specificity for discriminating NMIBC and MIBC, respectively (Table 2). Four-titer grouping based on the levels of urine Nectin-2/Nectin-4 (low/high, high/high, low/low, and high/low) showed a high discrimination capability for NMIBC and MIBC (Figure 3).

### 3.3. Prognostic Performance of Urine Nectin-2 and Nectin-4

In the NMIBC cohort with a median follow-up of 93 months (IQR 52–109), Kaplan–Meier curves for RFS, PFS, and OS were generated to evaluate the potential prognostic value of urine nectins. The cutoff values of the biomarkers for prognostic performance were determined using medians. There was a marginal association between high urine Nectin-2 and short RFS after TURBT, while there was no association of high urine Nectin-2 with PFS and OS (Figure 4A). Urine Nectin-4 expression was not associated with any outcomes (Figure 4B). In the cohort of 44 patients with MIBC with a median follow-up of 34 months (IQR 11–78), there was a marginal association between high urine Nectin-2 and short DFS after radical cystectomy, while there was no association of high urine Nectin-2 with CSS and OS (Figure 5A). Similar to the results in the NMIBC cohort, urine Nectin-4 was not associated with any of the outcomes (Figure 5B).

### 3.4. Correlation among Urine Nectins, Serum Nectins, and Tumor Nectins in MIBC

Tumor expression of targeted proteins, such as Nectin-4, is vital for ensuring antitumor responses to antibody–drug conjugates, such as EV [24,25]. Testing the tumor expression levels of nectins using IHC staining analysis is time-consuming and low-cost. Therefore, to investigate whether the tumor expression levels of nectins could be predicted using urine and serum nectins, we performed a correlation analysis among urine, serum, and tumor nectins in the MIBC cohort. There was no significant correlation between the levels of urine Nectin-2, serum Nectin-2, and tumor Nectin-2 (Figure 6A–C). The level of urine Nectin-4, though not serum Nectin-4, was slightly correlated with the tumor H-score of Nectin-4 (Spearman’s *ρ* = 0.31, *p* = 0.044). In addition, there was a positive correlation between urine Nectin-4 and serum Nectin-4 (Spearman *ρ* = 0.33, *p* = 0.027).

## 4. Discussion

The present study suggested the potential clinical utility of urine Nectin-2 and Nectin-4 as non-invasive biomarkers in the management of BCa. Non-invasive urine tests for early detection and postoperative surveillance are required for both patients and the healthcare system. To overcome the low sensitivity of urine cytology, several urine-based tests have been developed and approved for the clinical management of BCa. UroVysion is a DNA-based molecular cytology for BCa that utilizes multicolor fluorescent in situ hybridization probes for the detection of aneuploidy in chromosomes 3, 7, and 17, and the loss of the 9p21 locus, in which p16, the tumor suppressor gene, resides [1,26]. UroVysion is intended for use as an aid in patients with hematuria suspected of having BCa, and in those with a history of UC undergoing surveillance for tumor recurrence. The diagnostic accuracy of UroVysion is superior to that of cytology, with a reported overall sensitivity ranging from 55 to 81% [26]. However, the relatively low specificity (with reported overall sensitivity ranging from 65 to 96%) and poor cost performance hinder the widespread use of this non-invasive urine test. In this study, urine Nectin-4 exhibited 98% sensitivity, which appears to outperform other urine-based tests [27] and 80% specificity. Another advantage of urinary nectin testing is the simplicity of the enzyme immunoassay-based method. Some FDA-approved urine tests, such as BTA TRAK™/BTA stat^®^ and NMP22 BC^®^/NMP22^®^ BladderChek^®^ Test, share ELISA-based quantitative measurement and point-of-care qualitative measurement [28]. Similar to these commercially available assays, the development of point-of-care assay kits might spread clinical and at-home use of urine Nectin-4 assay to predict the risk of BCa.

Reducing the frequency of surveillance cystoscopy in NMIBC, especially in non-high-risk NMBIC, which protects patients from potential discomfort, anxiety, the risk of urinary tract infection, and the burden of medical costs without compromising detection rates, would be of great interest to clinicians and patients. Recent advances in mRNA-based urine tests, such as Cxbladder, have provided high sensitivity and negative predictive value for detecting recurrent UC [29]. Koya et al. validated the clinical utility of a new surveillance protocol that incorporated the Cxbladder Monitor test in real-world practice [30]. Based on the results of samples from 309 patients, consisting of 257 (83%) low-risk and 52 (17%) high-risk patients with NMIBC, the Cxbladder Monitor test accurately identified a high proportion of patients (78%) who were safely managed with only one cystoscopy per year. Because our cohort included only primary cases of BCa, we did not evaluate the potential of urine nectin for BCa monitoring. Further prospective studies are required to validate posttreatment surveillance protocols that incorporate urine nectin tests.

Previous studies have investigated the potential of Nectin-2 and Nectin-4 as biomarkers and treatment targets in various cancer types [10,11,12,13,14,15]. In ovarian cancer, serum Nectin-4 may serve as a potential diagnostic marker that helps discriminate benign gynecological diseases from malignancy in a panel with CA125 [10]. Another cohort study comprising 131 patients with ovarian cancer and 100 age-matched healthy controls demonstrated that preoperative collected serum Nectin-4 levels were significantly higher in patients with ovarian cancer than in controls [11]. Remarkably, a higher level of serum Nectin-4 was detected, especially in patients with early stage ovarian cancer, such as those with low CA125 [12]. In a breast cancer, Nectin-4 was predominantly overexpressed in ductal breast carcinoma tissues. Quantitative PCR analysis revealed that the mRNA expression of Nectin-4 was strongly correlated with the basal-like subtype and negatively correlated with the luminal-like subtype, which was not consistent with the previous results of UC analyses [25]. Additionally, serum Nectin-4 is a marker of disease progression and its level is correlated with the number of metastases [12]. A recent systematic review showed that a high serum Nectin-4 level correlated with treatment efficiency and disease progression [31]. However, there is limited evidence regarding the effects of urine nectins in patients with BCa.

Our unique investigation revealed the potential of urine nectins to discriminate between NMIBC and MIBC (Figure 3). Hoffman-Censits et al. conducted IHC analysis of BCa tissues to compare Nectin-4 expression according to tumor grade, tumor stage, and the presence of non-UC histotypes [9]. With a cutoff of 15 < H-score, 97% of non-invasive papillary tumors (Ta low-grade UC) and 80% of T1 tumors were positive for Nectin-4, while only 43% in MIBC cases were positive [32]. Given that nectins are exclusively involved in cell–cell adhesion, similar to cadherin proteins, it is reasonable that nectin expression is inversely correlated with epithelial–mesenchymal transition, which represents a high tumor grade and aggressiveness [16]. In the present study, the urine Nectin-2 was higher in MIBC than in NMIBC, whereas the urine Nectin-4 was higher in NMIBC than in MIBC (Figure 1A,C). Using the optimal cutoff values, four-titer grouping based on the level of urine Nectin-2/Nectin-4 (low/high, high/high, low/low, and high/low) demonstrated high diagnostic accuracy for muscle invasiveness (Figure 3).

Tumor expression of targeted proteins such as Nectin-4 is vital for obtaining antitumor responses to antibody–drug conjugates, such as EV [23,24]. Thus, tumor expression of Nectin-4 before EV treatment can be a useful predictive factor for preferable outcomes. A simple evaluation such as a liquid biopsy is appropriate for bedside management. In the present study, the nectin tumor H-score was correlated with urine nectin and serum nectin levels in patients with MIBC who were expected to receive EV treatment in the future. Only the level of urine Nectin-4, not serum Nectin-4, was correlated with the tumor H-score of Nectin-4, and there was a positive correlation between the level of urine Nectin-4 and serum Nectin-4 (Figure 6). This urine signature may be able to identify a subgroup of patients with a high probability of responding to EV treatment. However, only two patients were treated for EV in our cohort; therefore, the potential of urine and serum level of Nectin-4 as predictive biomarkers for EV treatment could not be evaluated. As our cohort did not include patients with metastatic UC, serum Nectin-4 might be unsuitable for correlation analysis between tumor expression and serum levels. Repeated monitoring of circulating Nectin-4 protein or Nectin-4 expression in circulating tumor cells might be a useful biomarker during EV treatment.

This study has some limitations. First, there was a significant difference in age between patients with BCa and healthy controls and a relatively small number of controls. The control cohort consisted of seven males (70%) and three females (30%), which was unmatched with the cohort of BCa (84% males and 16% females). Given the strong male/female bias in BCa prevalence, the unbalanced population rate of male/female might affect the results. In addition, the controls did not include urine samples from patients with non-neoplastic marginal disorders, such as hematuria, urinary infection, urinary inflammation, and benign bladder tumors. Second, we did not include all the consequent patients who were treated between January 2010 and December 2021 in our hospital. The inclusion criteria were set up, resulting in a total of 122 patients analyzed, which might cause potential selection bias. Third, we did not compare the diagnostic accuracy of urine nectins with that of other urine examinations, such as UroVysion and MRNA-based kits. Lastly, both Nectin-2 and Nectin-4 were not specific to the cancer tissues, which could impact on the results of this study.

## 5. Conclusions

Although the results of this study need to be externally validated, urine nectins may serve as novel diagnostic biomarkers for BCa. In addition, this easy-to-use urine signature may be able to discriminate between NMIBC and MIBC.

## Figures and Tables

**Figure 1 cancers-15-02565-f001:**
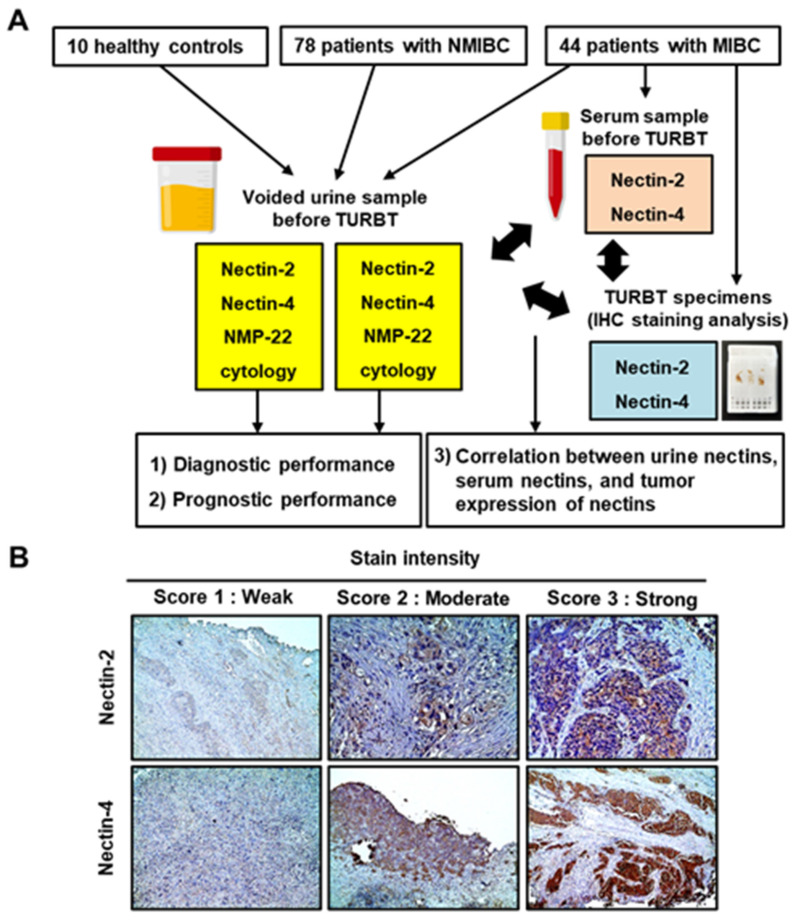
The study design and immunohistochemical staining analysis of Nectin-2 and Nectin-4. (**A**) Flowchart showing the study design. Study participants included 10 healthy controls, 78 patients with NMIBC, and 44 patients with MIBC. Pre-TURBT-voided urine samples were subjected to ELISA to measure Nectin-2, Nectin-4, and NMP-22 levels. Pre-TURBT serum samples were subjected to ELISA to measure Nectin-2 and Nectin-4. Paraffin-embedded formalin-fixed tissues from TURBT specimens were subjected to immunohistochemical staining analyses. (**B**) Representative images of the staining intensity (scores 1–3) of Nectin-2 and Nectin-4 in MIBC. Nectins are predominantly expressed in the membrane and the cytoplasm of urothelial carcinoma cells. Images were captured at ×100 or ×200 magnifications. Abbreviations: TURBT, transurethral resection of bladder tumor; NMIBC, non-muscle MIBC, muscle-invasive bladder cancer; IHC, immunohistochemistry.

**Figure 2 cancers-15-02565-f002:**
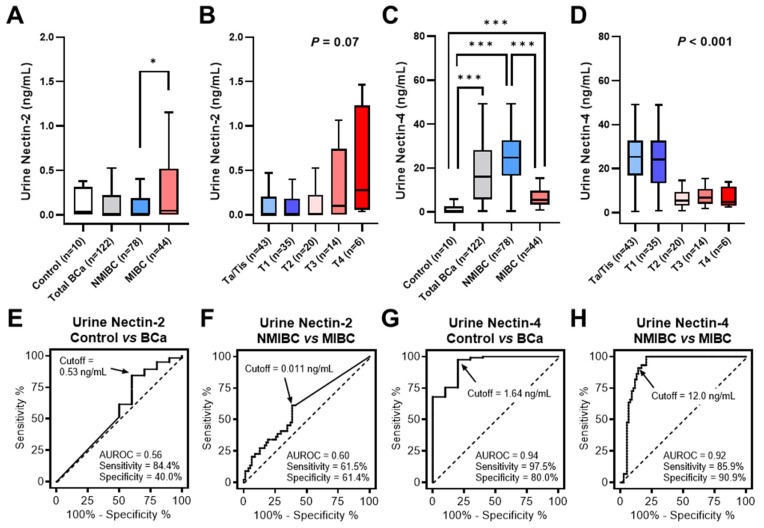
Diagnostic performance of urine Nectin-2 and Nectin-4. (**A**–**D**), The levels of urine Nectin-2 and Nectin-4 were compared among healthy controls, patients with BCa, those with NMIBC, and those with MIBC, or according to the T category. Data on urine nectin levels are visualized as box-and-whisker plots. The Mann–Whitney *U* test was used to compare the levels between the two groups. Two-sided Mann–Whitney *U* test and Kruskal–Wallis test were used to compare the levels of urine nectin between the two groups and among the four groups, respectively. Asterisks, * and *** indicate statistically significance, *p* < 0.05 and *p* < 0.001. (**E**–**H**), The diagnostic performance of urine Nectin-2 and Nectin-4 was evaluated using (ROC) curve analyses and the area under the receiver operating characteristic curve (AUROC) values, followed by determination of the optimal cut-off value by calculating the Youden index. Abbreviations: BCa, bladder cancer; NMIBC, non-muscle-invasive bladder cancer; MIBC, muscle-invasive bladder cancer.

**Figure 3 cancers-15-02565-f003:**
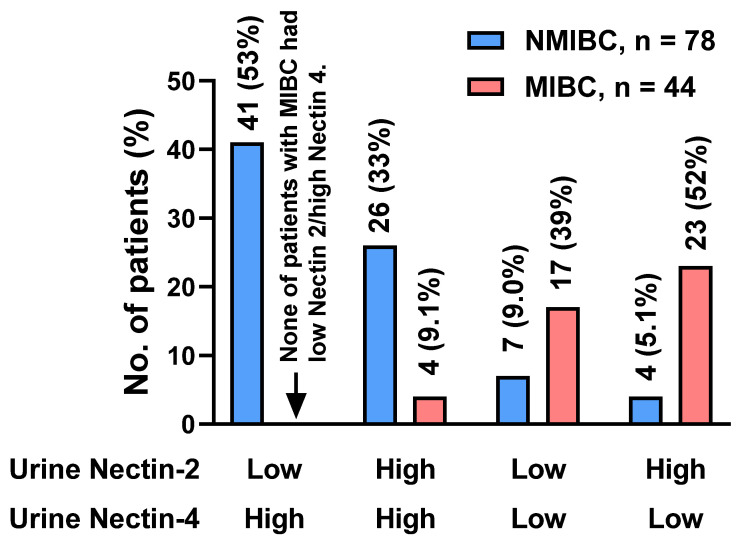
Discrimination capability for NMIBC and MIBC of urine Nectin-2 and Nectin-4. Based on receiver operating characteristic analysis, the optimal cutoff values for discriminating NMIBC from MIBC were 0.011 ng/mL for Nectin-2 and 12.0 ng/mL for Nectin-4. Four-titer grouping based on the levels of urine Nectin-2/Nectin-4 (low/high, high/high, low/low, and high/low) showed a high discrimination capability for NMIBC and MIBC. The vertical axis indicates the population and rate of each group. Abbreviations: NMIBC, non-muscle-invasive bladder cancer; MIBC, muscle-invasive bladder cancer.

**Figure 4 cancers-15-02565-f004:**
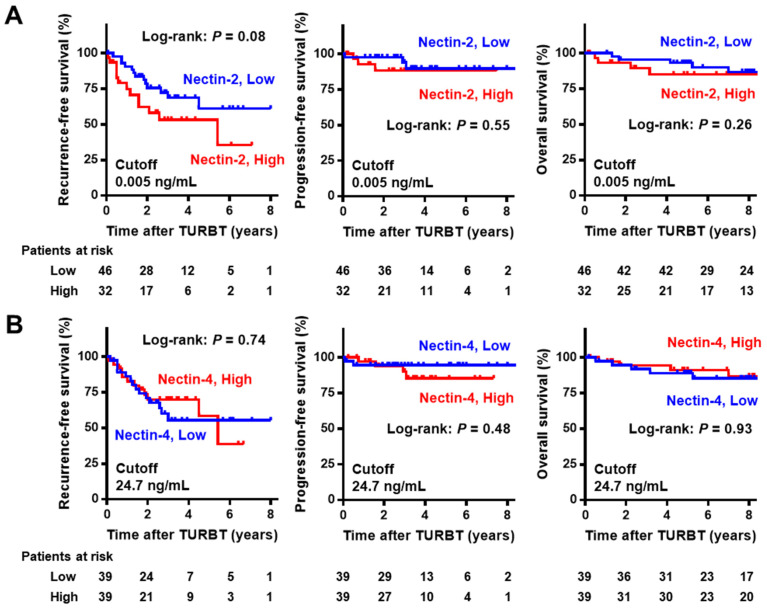
Survival curves after initial transurethral resection in non-muscle-invasive bladder cancer according to the level of Nectin-2 (**A**) and Nectin-4 (**B**). The cut-offs for the prognostic performance of urine Nectin-2 and Nectin-4 levels were determined using the median level. In the NMIBC cohort, intravesical recurrence-free survival, progression-free survival, and overall survival from the day of initial TURBT were obtained using the Kaplan–Meier method and compared using the log-rank test. Abbreviations: NMIBC, non-muscle-invasive bladder cancer; MIBC, muscle-invasive bladder cancer; TURBT, transurethral resection of bladder tumors.

**Figure 5 cancers-15-02565-f005:**
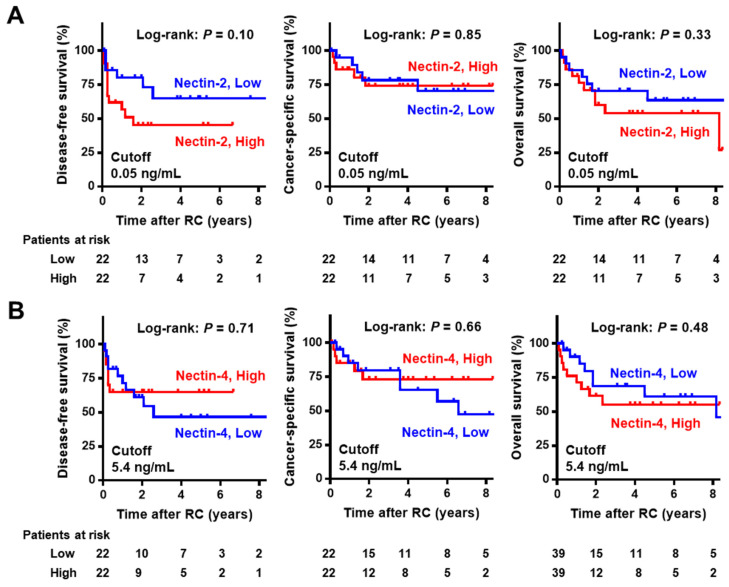
Survival curves after radical cystectomy in muscle-invasive bladder cancer according to the level of Nectin-2 (**A**) and Nectin-4 (**B**). The cut-offs for the prognostic performance of urine Nectin-2 and Nectin-4 levels were determined using the median level. In the MIBC cohort, disease-free survival, cancer-specific survival, and overall survival from the day of radical cystectomy were obtained using the Kaplan–Meier method and compared using the log-rank test. Abbreviations: NMIBC, non-muscle-invasive bladder cancer; MIBC, muscle-invasive bladder cancer; RC, radical cystectomy.

**Figure 6 cancers-15-02565-f006:**
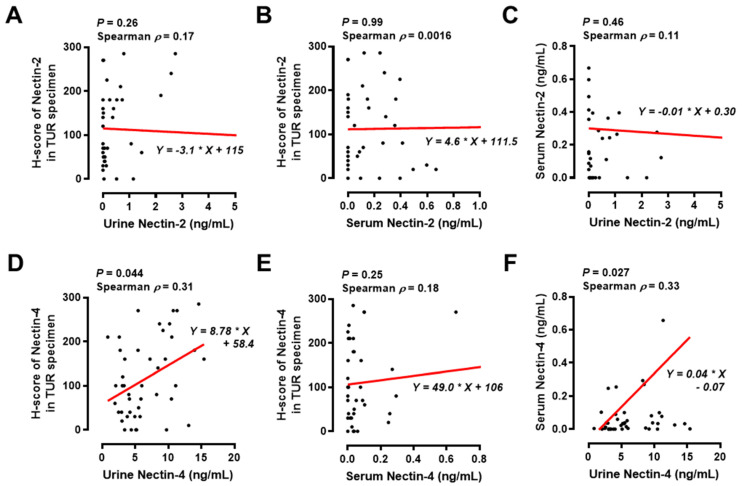
Correlation analysis among urine nectin, serum nectin, and tumor nectin levels in the MIBC cohort. The correlation between urine Nectin-2 and tumor H-score of Nectin-2 (**A**), between serum Nectin-2 and tumor H-score of Nectin-2 (**B**), between urine Nectin-2 and serum Nectin-2 (**C**), between urine Nectin-4 and tumor H-score of Nectin-4 (**D**), between serum Nectin-4 and tumor H-score of Nectin-4 (**E**), and between urine Nectin-4 and serum Nectin-4 (**F**) were examined using Spearman’s correlation and linear regression analysis; *p* values and Spearman *ρ* values are shown in figures. The absolute values of Spearman *ρ* < 0.2–0.4 were considered to indicate slightly correlated. The asterisk in the formula is the multiplication sign. Abbreviations: TUR, transurethral resection.

**Table 1 cancers-15-02565-t001:** Association between baseline characteristics and pre-TURBT urine nectins in 122 patients with bladder cancer.

Variables		No. of Cases	Urine Nectin-2 (ng/mL)			Urine Nectin-4 (ng/mL)		
			Median (IQR)	Mean ± SEM	*p* Value	Median (IQR)	Mean ± SEM	*p* Value
No. of cases		122 (100%)	0.0 (0.0–0.2)	0.40 ± 0.13	-	16.0 (5.8–28.1)	18.3 ± 1.2	-
Sex	Male	103 (84%)	0.0 (0.0–0.26)	0.43 ± 0.15	0.99 †	15.9 (5.5–28.8)	18.2 ± 1.4	0.75 †
	Female	19 (16%)	0.02 (0.0–0.21)	0.21 ± 0.09		16.6 (10.5–26.4)	18.6 ± 3.0	
Age	70 yo or less	51 (42%)	0.0 (0.0–0.15)	0.17 ± 0.05	0.052 †	20.4 (9.6–30.3)	20.5 ± 1.8	0.06 †
	71 yo or more	71 (58%)	0.04 (0.0–0.40)	0.56 ± 0.22		13.1 (5.2–25.3)	16.6 ± 1.7	
Urine cytology	Negative	61 (50%)	0.0 (0.0–0.21)	3.0 ± 0.09	0.47 †	18.5 (8.9–27.4)	18.7 ± 1.5	0.42 †
	Positive	61 (50%)	0.03 (0.0–0.36)	5.0 ± 0.25		12.8 (5.4–29.9)	17.8 ± 1.9	
Urine NMP-22 #	Negative (£10 U/mL)	59 (48%)	0.0 (0.0–0.15)	0.18 ± 0.08	0.002 †	17.4 (5.9–27.9)	18.5 ± 1.7	0.72 †
	Positive (>10 U/mL)	63 (52%)	0.06 (0.0–0.53)	0.60 ± 0.24		13.4 (5.7–29.8)	18.0 ± 1.8	
Tumor grade (WHO2004)	Low grade	41 (34%)	0.0 (0.0–0.21)	0.26 ± 0.11	0.64 †	0	24.0 ± 1.6	<0.001 †
	High grade	81 (66%)	0.0 (0.0–0.38)	0.46 ± 0.19		9.7 (4.3–24.2)	15.4 ± 1.6	
Carcinoma in situ	Absent	75 (61%)	0.03 (0.0–0.40)	0.55 ± 0.21	0.03 †	16.6 (8.2–27.5)	18.7 ± 1.6	0.59 †
	Present	47 (39%)	0.0 (0.0–0.14)	0.15 ± 0.05		10.5 (5.4–30.3)	17.5 ± 2.0	
Clinical T category #*#*	Ta/Tis	43 (35%)	0.0 (0.0–0.20)	0.25 ± 0.11	0.07 ‡	25.3 (16.9–32.7)	24.9 ± 1.8	<0.001 ‡
	T1	35 (29%)	0.0 (0.0–0.18)	0.19 ± 0.07		24.1 (13.4–32.7)	24.9 ± 2.4	
	T2	24 (20%)	0.0 (0.0–0.23)	0.90 ± 0.62		5.4 (3.1–9.5)	6.1 ± 0.7	
	T3	14 (11%)	0.10 (0.0–0.74)	0.44 ± 0.17		6.7 (3.9–10.7)	7.2 ± 1.1	
	T4	6 (5%)	0.28 (0.05–1.23)	0.55 ± 0.25		4.7 (3.1–11.8)	6.7 ± 1.9	
Muscle invasiveness	NMIBC (Ta/Tis/T1)	78 (64%)	0.0 (0.0–0.19)	0.22 ± 0.07	0.04 †	24.7 (16.6–32.7)	24.9 ± 1.4	<0.001 †
	MIBC (T2-4)	44 (36%)	0.05 (0.0–0.52)	0.70 ± 0.34		5.4 (3.3–9.7)	6.5 ± 0.6	

TURBT, transurethral resection; IQR, interquartile range; SEM, standard error of the mean; # The cutoff value (10 U/mL) is based on FDA recommendation and published literatures ##, 8th Edition of the UICC TNM classification; †, Mann–Whitney U test; ‡, Kruskal-Wallis test.

**Table 2 cancers-15-02565-t002:** Performance characteristics of urine markers in pre-TURBT voided urine.

Performance Characteristics	Urine Nectin-2		Urine Nectin-4		NMP-22	VUC
	Control vs. BCa	NMIBC vs. MIBC	Control vs. BCa	NMIBC vs. MIBC	Control vs. BCa	Control vs. BCa
AUROC	0.56	0.60	0.94	0.92	-	-
Cutoff value	0.53 ng/mL †	0.011 ng/mL †	1.64 ng/mL †	12.0 ng/mL †	10 U/mL ‡	-
Sensitivity (%)	84 (77–90)	61 (46–74)	98 (93–99)	91 (79–96)	52 (43–60)	47 (38–56)
Specificity (%)	40 (17–69)	61 (50–72)	80 (49–96)	86 (76–92)	100 (72–100)	100 (72–100)
Positive predictive value (%)	95 (89–97)	47 (35–60)	98 (94–99)	78 (65–88)	100 (94–100)	100 (94–100)
Negative predictive value (%)	17 (7.0–37)	74 (52–83)	73 (43–90)	94 (86–98)	14 (8.1–25)	13 (7.4–23)
Accuracy (%)	81	57	96	81	55	51
McNemar test, *p* value for comparison with VUC	*p* < 0.001	-	*p* < 0.001	-	*p* = 0.50	-

†, Mann–Whitney U test; ‡, Kruskal-Wallis test.

## Data Availability

The data presented in this study are available upon request from the corresponding author (M.M.).

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
