# Peer review of "Diagnostic and Prognostic Roles of Urine Nectin-2 and Nectin-4 in Human Bladder Cancer"

_cancers, 2023, doi:10.3390/cancers15092565_

Round 1

Reviewer 1 Report

Dear authors,

Thank you for submitting your work! It contains interesting experiments testing tumor nectin expression in serum, urine, and tissue. The presence of nectines in the urine may be predictive and informative for discriminating NMIBC and MIBC but lacks prognostic value.

My comments concerning your work are:

1)    Introduction: Please include your study hypothesis and underlying rationale.

2)    Patients and Methods: Urine samples from 122 patients and 10 healthy controls were analyzed, even though the studied period was >10 years; which may cause selection bias. How do the authors explain this small sample sizes – how were these individuals selected?

Also, elaborate on the study design (pro- or retrospective) and include the IRB number.

Concerning the processing of voided urine, serum and tissue samples, it is not clear if this was done in all groups (healthy, NMIBC, and MIBC), please clarify. If not, why was this not performed consistently?

In the statistical analysis, the authors describe the optimal cut-off value for urine nectines using them as diagnostic markers was defined with the Youden Index. However, when using these markers for prognosis, the cut-off was the median. Please explain the different approach for the statistical analysis of your markers of interest.

3)    Results: The table on top of page 6 includes “Clinical T category“ and “Muscle invasiveness“. Please choose one of these variables as they are similar and reflect each other.

In the section describing the prognostic performance of urine Nectin-2 and Nectin-4, convert the median follow-up time from months to years, to ease readability.

4)    Discussion: Please mention in the limitation section of the discussion the low tissue and cancer specificity of Nectin-2 and the low cancer specificity of Nectin-4; and the impact of this in the context of the study results. 

5) Conclusion: Please revise the sentence "In addition, this easy-to-use urine signature may be able to discriminate between NMIBC and MIBC and identify a subgroup of 372 patients with a high probability of responding to EV treatment", as only two patients were treated with EV and you did not formal response evaluation. But you may mention this comment in the discussion.

Author Response

1)    Introduction: Please include your study hypothesis and underlying rationale.

>reply

Thanks for the recommendation. The manuscript lacked study hypothesis and rationale. We have added the sentences in the Introduction section (Page 2, lines 69-74) as shown below: ‘Our recent study demonstrated that nectins are predominantly expressed in the mem-brane and/or cytoplasm of UC cells with no or faint expression of Nectin-1, slight ex-pression of Nectin-3 and high expression of Nectin-2 and Nectin-4 [16]. To the best of our knowledge, no study has addressed the potential diagnostic and prognostic values of urine Nectin-2 and Nectin-4 in any malignancy. We hypothesized that urine Nec-tin-2 and Nectin-4 were significantly higher in patients with BCa as compared to healthy controls.’

The reviewer’s understanding would be appreciated.

2)    Patients and Methods: Urine samples from 122 patients and 10 healthy controls were analyzed, even though the studied period was >10 years; which may cause selection bias. How do the authors explain this small sample sizes – how were these individuals selected?

>reply

This is a great comment. Exactly, much more than 122 patients with BCa underwent treatment between January 2010 and December 2021 in our hospital. Inclusion criteria of this study was 1) pre-TURBT voided urine was preserved properly and available and 2) written informed consent was obtained. Meeting the inclusion criteria, the sample size turned out to be 122 patients. I have added the patient selection process and inclusion criteria in the Patients and Methods section (Page 2, lines 87-93) as follows: ‘The inclusion criteria of this study were 1) pre-TURBT voided urine was preserved properly and available for ELISA (for NMIBC), 2) all of the pre-TURBT voided urine, pre-TURBT serum, and paraffin-embedded, formalin-fixed tissue blocks of TURBT specimens were preserved properly and available for the analysis (for NMIBC), and 3) written informed consent was obtained. A total of pre-TURBT-voided urine samples from 78 patients with NMIBC and 44 patients with MIBC and ten healthy controls were analyzed by ELISA.’ A comment regarding potential selection bias was added in the Limitation section (Page 13. Line 370-374) as follows: ‘we did not include all the consequent patients who were treated between January 2010 and December 2021 in our hospital. The inclusion criteria were set up, resulting in a total of 122 patients analysed, that might cause potential selection bias.’

The reviewer’s understanding would be appreciated.

3)    Also, elaborate on the study design (pro- or retrospective) and include the IRB number.

>reply

Thanks for the comment. We have added ‘retrospective’ in the first line of the paragraph of 2.1. Patient selection and data collection (Page 2, line 84). Plus, we wrote the description (Page 13, line 404-405) as follows: ‘Institutional Review Board Statement: The Ethics Committee of the Nara Medical University approved this study (reference number: 1256). Patient anonymity was preserved.’

4)   Concerning the processing of voided urine, serum and tissue samples, it is not clear if this was done in all groups (healthy, NMIBC, and MIBC), please clarify. If not, why was this not performed consistently?

>reply

This is an important concern. Our study is retrospective study using the samples which was collected before and preserved properly. All the ELISA were performed at the same time. We have added a sentence in 2.3. Measurement of nectins in urine and serum (Page 3, lines 124-125) as follows: ‘To avoid possible consistency among assays, all the ELISA were performed at the same time.’

The reviewer’s understanding would be appreciated.

5)   In the statistical analysis, the authors describe the optimal cut-off value for urine nectines using them as diagnostic markers was defined with the Youden Index. However, when using these markers for prognosis, the cut-off was the median. Please explain the different approach for the statistical analysis of your markers of interest.

>reply

We understand the reviewer’s concern. Youden index has been conventionally and most frequently used approach for defining the best cutoff values diagnosing the disease morbidity. However, the same cutoff values sometimes did not work to compare the survival. Another thing is that the cutoff values defined with Youden index do not divide the cohort evenly into two groups (for example, low expression and high expression). Low number of patients often cause low number of events, which is associated with low statistical power in comparison. Therefore, we have decided to use median value for dividing the patient cohort evenly into two groups.

The reviewer’s understanding would be appreciated.

6)    Results: The table on top of page 6 includes “Clinical T category“ and “Muscle invasiveness“. Please choose one of these variables as they are similar and reflect each other.

>reply

Thanks for the recommendation. Exactly, “Clinical T category“ and “Muscle invasiveness“ overlapped each other. However, one of our interests was comparing urine nectins between NMIBC and MIBC. The analysis with the cutoff between T1 and T2 would help the leaders’ better understanding. We really appreciate the reviewer’s understanding if the current data presentation could be kept as they stand.

7)    In the section describing the prognostic performance of urine Nectin-2 and Nectin-4, convert the median follow-up time from months to years, to ease readability.

>reply

Thanks for the recommendation. We have converted the median follow-up time from months to years.

8)    Discussion: Please mention in the limitation section of the discussion the low tissue and cancer specificity of Nectin-2 and the low cancer specificity of Nectin-4; and the impact of this in the context of the study results.

Thanks for the recommendation. We have added the sentence in the limitation section.

The reviewer’s understanding would be appreciated.

8) Conclusion: Please revise the sentence "In addition, this easy-to-use urine signature may be able to discriminate between NMIBC and MIBC and identify a subgroup of patients with a high probability of responding to EV treatment", as only two patients were treated with EV and you did not formal response evaluation. But you may mention this comment in the discussion.

>reply

Thanks for this helpful comment. We totally agree with the reviewer’s comment. We re-wrote the Discussion and Conclusion sections as the reviewer recommended.

Reviewer 2 Report

The manuscript by Miyake et al. describes a study aimed at determining the potential diagnostic and prognostic value of urine levels of the cell adhesion proteins nectin 2 and nectin 4 in a cohort of 78 patients with noninvasive bladder cancer, 44 with muscle-invasive disease, and 10 controls. In general, nectin 4 was more abundant than nectin 2 in the patients’ urine samples. Urine nectin 2 levels were higher in MIBC than in NMIBC; nectin 4 showed the opposite trend. As diagnostic markers, urine nectin 2 and nectin 4 were each more sensitive, but considerably less specific, than urine NMP-22 or cytology. Four-way evaluation of urine nectin 2/nectin 4 values (High-Low) discriminated between noninvasive and invasive bladder cancer. Urine levels of the 2 nectins were not associated with prognosis (recurrence-free, progression-free, overall survival). The Authors propose nectin 2 and nectin 4 as diagnostic biomarkers for bladder cancer, and the possible utility of urine nectin 4 as a marker to select or monitor patients treated with the anti-nectin 4-monomethyl auristatin E conjugate enfortumab-vedotin.

The topic of the study and results are relevant, and the manuscript is clearly written.

The following points should be addressed.

On lines 22-24 of the Simple Summary, the Authors mention the nectin signature for identifying patients likely to respond to enfortumab-vedotin. They do not provide data to support this statement.

As pointed out by the Authors on lines 360-364, the study included a very small number of controls who were not age-matched. The numbers of male and female controls should also be provided, as this might represent an additional limitation to the study, given the strong male/female bias in bladder cancer prevalence.

Were the nectins detected in the urine of the controls? If so, median and mean levels in males and females should be reported in Table 1. If the nectins were undetectable, this should be stated in the text.

On lines 81-62, the Authors should provide a brief description of NMP-22 and cite Landman J et al. Urology 1998;52:398-402, which describes its elevated expression in urothelial cancers and presence in urine.

On line 114, the text should state ‘Alere NMP22 BladderChek Test’

On line 206, the text states that ‘2 of 10 controls (20%) were negative’. Did the Authors mean to write ‘positive’?

In Figure 3, the arrow - No MIBC label is not easy to understand. The legend should state that none of the MIBC patients had low nectin 2/high nectin 4.

On line 234, the text should read …between nectin 2 levels and PFS or OS…

On lines 237-238, the text should read …between nectin 2 levels and CSS or OS…

The text in Section 3.3 should define the prognosis abbreviations (RFS, PFS, OS, DFS, CSS).

On line 262, did the Authors mean to state that IHC is costly (not low-cost)?

Figure 6 contains some labeling errors- replace Speaman with Spearman (all panels) and Serume with Serum (Panel B).

On line 281, the Authors state that the study confirmed the clinical utility of the nectin markers. This statement is too strong. The study indicated the potential clinical utility of the nectins.

On line 304, the text should read …urinary tract infection…

Author Response

1) On lines 22-24 of the Simple Summary, the Authors mention the nectin signature for identifying patients likely to respond to enfortumab-vedotin. They do not provide data to support this statement.

>reply

We totally agree with the reviewer’s comment. This is just our speculation and there is no evidence to support the speculation. We have decided to delete the sentence. Thanks.

2) As pointed out by the Authors on lines 360-364, the study included a very small number of controls who were not age-matched. The numbers of male and female controls should also be provided, as this might represent an additional limitation to the study, given the strong male/female bias in bladder cancer prevalence.

>reply

Thanks for the comment. Exactly. Small number of controls and is one of the biggest limitations of this study. The control cohort consisted of seven males (70%) and three females (30%), which was a bit unmatched compared to the cohort of BCa, 84% males and 16% females. We have added a couple of sentences in the Patents and Methods section and the limitation section.

We appreciate the reviewer’s recommendation.

3) Were the nectins detected in the urine of the controls? If so, median and mean levels in males and females should be reported in Table 1. If the nectins were undetectable, this should be stated in the text.

>reply

Thanks for the comment. Yes, nectins were detected in urine samples of the control cohort. Table 1 focuses on the data of patients with BCa. It might be confusing to add the data of the control group in the Table 1. Instead, we have added the data of urine nectins of the controls in the Results section (Page 5, lines 189-191) as follows: ‘The mean mean ± standard error of the mean of urine Nectin-2 and Nectin-4 in the control group were 0.63 ± 0.34 ng/mL and 1.68 ± 1.01 ng/mL, respectively.’

We appreciate the reviewer’s recommendation.

4) On lines 81-62, the Authors should provide a brief description of NMP-22 and cite Landman J et al. Urology 1998;52:398-402, which describes its elevated expression in urothelial cancers and presence in urine.

>reply

Thanks for the comment. A brief description regarding NMP22 would be vital to make better understanding of the comparators. We have added two citations and description in the 2.3. Measurement of nectins in urine and serum sections (Page 3, lines 124-) as follows: ‘The NMP-22 assays are FDA approved for the detection and surveillance of BCa in urine samples. For BCa detection, urinary NMP-22 tests have diagnostic sensitivities ranging from 47–100% and specificities ranging from 55–98% [20,21].’

  1. Landman J et al. Urology 1998;52:398-402

https://pubmed.ncbi.nlm.nih.gov/9730450/

  1. Sözen, S. et al. Eur Urol. 1999 Sep;36(3):225-9.

https://pubmed.ncbi.nlm.nih.gov/10450007/

  1. Miyake M, et al. BMC Urol. 2012 Aug 28;12:23.

https://pubmed.ncbi.nlm.nih.gov/22928931/

5) On line 114, the text should state ‘Alere NMP22 BladderChek Test’

>reply

Thanks for the comment. the NMP22 BladderChek is a point-of-care qualifying methods. To make sure, we have revised the official name of the kit as following: Alere NMP22® Bladder Cancer ELISA Test.

6) On line 206, the text states that ‘2 of 10 controls (20%) were negative’. Did the Authors mean to write ‘positive’?

>reply

Thanks for the comment. Exactly. We have changed ‘negative’ to ‘positive’.

7) In Figure 3, the arrow - No MIBC label is not easy to understand. The legend should state that none of the MIBC patients had low nectin 2/high nectin 4.

>reply

Thanks for the helpful comment. Exactly. It seemed to be hard to understand ‘No MIBC’. We have revised the Figure 3. It shows ‘None of patients with MIBC had low Nectin2/high Nectin 4.’

8) On line 234, the text should read …between nectin 2 levels and PFS or OS…

>reply

Thanks for the comment. Exactly, this is weird. We have revised the sentence as follows: ‘while there was no association of high urine Nectin-2 with PFS and OS’

9) On lines 237-238, the text should read …between nectin 2 levels and CSS or OS…

>reply

Thanks for the comment. Exactly, this is weird. We have revised the sentence as follows: ‘whereas there was no association of high urine Nectin-2 with CSS and OS’

10) The text in Section 3.3 should define the prognosis abbreviations (RFS, PFS, OS, DFS, CSS).

>reply

Thanks for the comment. It is important to define the terminology. We have added the description regarding the definition in 2.5. Statistical analysis section of the Patients and Methods (Page 5, lines 170-176).

11) On line 262, did the Authors mean to state that IHC is costly (not low-cost)?

>reply

Thanks for the comment. This is weird. This is just a typo. We have revised to ‘high-cost’.

10) Figure 6 contains some labeling errors- replace Speaman with Spearman (all panels) and Serume with Serum (Panel B).

>reply

Thanks for the comment. We have modified the labelling.

11) On line 281, the Authors state that the study confirmed the clinical utility of the nectin markers. This statement is too strong. The study indicated the potential clinical utility of the nectins.

>reply

Thanks for the comment. We totally agree with the reviewer’s idea. Thus, we have modified ‘confirmed the clinical utility’ to ‘suggested the potential clinical utility’.

We appreciate the reviewer’s recommendation.

12) On line 304, the text should read …urinary tract infection…

>reply

Thanks for the comment. We have modified it to ‘urinary tract infection’.

Round 2

Reviewer 1 Report

Dear authors,

Thank you for addressing the comments!